# Occurrence of Microplastics in Tap and Bottled Water: Current Knowledge

**DOI:** 10.3390/ijerph19095283

**Published:** 2022-04-26

**Authors:** Isabella Gambino, Francesco Bagordo, Tiziana Grassi, Alessandra Panico, Antonella De Donno

**Affiliations:** Department of Biological and Environmental Science and Technology, University of Salento, 73100 Lecce, Italy; isabella.gambino@unisalento.it (I.G.); francesco.bagordo@unisalento.it (F.B.); alessandra.panico@unisalento.it (A.P.); antonella.dedonno@unisalento.it (A.D.D.)

**Keywords:** microplastics, drinking water, tap water, bottled water, exposure, toxicity

## Abstract

A narrative review was carried out to describe the current knowledge related to the occurrence of MPs in drinking water. The reviewed studies (*n* = 21) showed the presence of microplastics (MPs) in tap (TW) and bottled (BW) water, increasing concerns for public health due to the possible toxicity associated with their polymeric composition, additives, and other compounds or microorganism adsorbed on their surface. The MP concentration increase by decreasing particles size and was higher in BW than in TW. Among BW, reusable PET and glass bottles showed a higher MP contamination than other packages. The lower MP abundance in TW than in natural sources indicates a high removal rate of MPs in drinking water treatment plants. This evidence should encourage the consumers to drink TW instead of BW, in order to limit their exposure to MPS and produce less plastic waste. The high variability in the results makes it difficult to compare the findings of different studies and build up a general hypothesis on human health risk. A globally shared protocol is needed to harmonize results also in view of the monitoring plans for the emerging contaminants, including MPs, introduced by the new European regulation.

## 1. Introduction

### 1.1. Microplastics in the Environment

Plastics are materials made up of long-chain polymers that are widely used in all sectors, including health and food, due to their lightness, strong plasticity and flexibility, thermal and electrical insulation, chemical resistance, durability and low cost. Their global production has rapidly and hugely increased over the last few decades reaching 359 million tons in 2018 [1] and posing serious problems of management when plastic products turn into waste at the end of their service life. Because of improper disposal practices, a large amount of plastic waste enters the environment every year causing numerous concerns about its impact on the quality of natural resources and the health of ecosystems [2,3].

In 2020, worldwide plastic production reached 367 million metric tons, resulting in 29.1 Mt of plastic waste [4,5]. According to Geyer et al. [3], 9% of plastic waste was recycled, 12% incinerated and the rest landfilled or dumped in the environment. According to Lebreton and Andrady [6], 60–99 million tons of plastics were inappropriately disposed of in the environment worldwide in 2015.

In the environment, plastic waste is subjected to biotic (i.e., biodegradation performed by microbes) and abiotic degradation (photodegradation due to UV exposure or weathering degradation as the result of waves and winds action) [7,8]. These processes lead to the formation of smaller plastic fragments which are considered microplastics (MPs). MPs are defined as particles < 5 mm in length [9]. Officially there are no lower limits for MPs, however, according to Gigault et al. [10], the lower size limit of MPs was set to 1 μm while plastic particles smaller than 1 μm were usually considered nanoplastics (NPs) [11] (Figure 1).

MPs can be divided into two large groups: primary MPs, intentionally produced, such as components of industrial or commercial products (paints, adhesives coatings, microbeads in cleansers and in cosmetics) and secondary MPs, resulting from the breakdown of larger aged-plastic debris [13]. Most commonly found MPs are made of Polyethylene terephthalate (PET), Polyethylene (PE) (Low-Density PE, Linear-Low-Density PE, High-Density PE), Polypropylene (PP), Polystyrene (PS), Polyvinyl Chloride (PVC), Polylactic Acid (PLA), Polyamide (PA), Polycarbonate (PC), Polyurethane (PU), Acrylonitrile butadiene styrene (ABS) [14].

MPs can be transported by wind and water flow due to their light weight [15]. Precipitations, surface runoff, infiltration, and riverine transport could be the main routes that transfer plastics from land to water [16,17].

To date, MPs are ubiquitous particles and have been found both in aquatic [18,19] and terrestrial [20,21] ecosystems becoming a global environmental issue. They have been studied and well documented in marine waters [22], surface water bodies [23,24], wastewater [25,26] and groundwater [27]. An increasingly substantial literature has demonstrated the ingestion and accumulation of MPs at any level of biological organization [28,29], indicating that they can be transported through the food chain even at the top trophic level [30,31,32].

### 1.2. Human Exposure to MPs

MPs are presently a pressing concern for public health too since they are present in all environmental matrices and in many products that people use daily. These tiny particles can enter the human body in different ways: dermal contact, airborne exposure and ingestion. Dermal contact represents the less significant exposure pathway and absorption across intact skin is unlikely because of the protective function of the corneum layer [33]. However, skin lesions might facilitate the penetration of small particles, or through catheters or syringes [34,35]. Airborne exposure represents an important pathway for MPs [36]. As demonstrated for other particles, they can reach bronchial lung tissues by inhalation leading to inflammation events [34,37].

Ingestion through contaminated foods represents one of the main pathways through which MPs enter the human body [38]. They were traced in many foods, such as fish products [39,40], edible mollusks [30], fruit and vegetables [41], table salt [42,43] and industrial and packaged food [44,45] as a result of environmental or process contamination. The ingested MPs are mostly excreted (up to 90%) [39] and were detected in humans’ feces [46] but the pathophysiological consequences of the amount of MPs retained in the human body are yet unclear.

### 1.3. Toxicity

The main evidence concerning the toxicity of MPs has resulted from studies carried out with animal models and in vitro experiments on human cells. They suggest the possible hazards due to these particles when they are ingested by humans [37,47,48].

In general, the effects derived from MPs could be classified as physical and chemical effects [37]. The former is related to the presence of the particles themselves in the organism, as well as their accumulation, and is mainly due to the particle size, shape and concentration. The latter is related to the polymeric composition, the leaching of chemical additives, and the presence of toxic compounds which can be found adsorbed to the external surface of MPs.

Studies on toxicity due to the polymeric composition of MPs in marine, freshwater, and terrestrial organisms reported various toxic effects but the most frequently mentioned were oxidative stress-related endpoints. There is a lack of toxicity data for humans in vivo at the moment. However, several studies have looked at the effect of pristine MPs in human cells in culture [49]. In these studies, as in animal models, the main detected endpoint was oxidative stress with reactive oxygen species (ROS) generation [50,51,52,53,54]. Other toxic effects regarded lipid metabolism, microbiota, neurotoxicity, inflammatory and immunological responses, cytotoxic effects, disruption of mitochondrial membrane potential, inhibition of plasma membrane ATP-binding cassette (ABC) transporter activity [47,52,55].

The toxicity of MPs was often attributed to the numerous additives added to the plastic polymers during the processing and fabrication of products either to modify their properties or to control their degradation. The most commonly used additives in different types of polymeric materials are: plasticizers (i.e., phthalates, adipates, etc.), flame retardants (i.e., PBDEs, antimony, etc.), stabilizers and antioxidants (i.e., BPA, cadmium and lead compounds, nonylphenol compounds, BHA, etc.), acid scavengers, light and heat stabilizers, lubricants, colorants and pigments, antistatic agents, slip compounds and thermal stabilizers [56]. Plastic additives were reported to be responsible for oxidative stress, endocrine disruption, impaired lipid metabolism and even cancer in many organisms, including humans [57,58].

MPs are able to absorb and concentrate on their surface many pollutants such as persistent organic pollutants (POPs), heavy metals, polychlorinated biphenyls (PCBs), antibiotics, endocrines disrupting chemicals (EDCs), bacteria, viruses, and resting stages of potentially hazardous organisms [59,60,61,62,63].

Dimension and shape as well as target cells/organs play a crucial role in exerting toxicity [49]. Size-related effects were reported in the literature and may influence the uptake across the epithelium and the translocation within the organisms into the lymphatic and circulatory systems [64].

The smaller the particle size, the higher the effects. The highest toxicity in human cell lines was registered when they were exposed to NPs, while larger particles (>100 μm) showed to have no significant effects in fishes [65]. Shape-dependent toxicity of MPs was observed in animal models. In general, fibers had a higher toxicity than fragments and spherical MPs [66,67]. Due to their tiny size, they escape water treatment processes so they could be detected in all water environments, including those used for drinking purposes [68]. Clear evidence of human exposure to MPs is given by their presence in human placenta [69], meconium samples [70], human feces [46], lung tissues [71], colon [72], nasal mucous [73], hair, hand skin, face skin, saliva [74] and blood [75]. At the human level, larger particles are expected to be excreted by feces while smaller particles could enter into the systemic circulation. However, at high concentrations or in the presence of toxic compounds, particles would likely infer alteration of viability or inflammation at the gut level. As demonstrated in mouse models, long-term accumulation of particles in liver tissues and chronic inflammation could lead to liver disease and metabolic problems, as well as accumulation in lung tissues could lead to chronic pulmonary disorders. Other effects could be associated with the chemical composition of MPs such as the presence of compounds considered biologically and toxicologically relevant to humans [49].

Finally, the effects could depend also on the time and intensity of exposure as well as the susceptibility of the host. In this regard, drinking water represents the most alarming issue considering the direct and long-term exposure of the entire population, including the most vulnerable groups such as children and sick people. Therefore, the occurrence of MPs both in tap and bottled water is now receiving more attention [36].

The purpose of this review was to describe the current knowledge related to the occurrence of MPs in drinking water through the analysis of the literature concerning tap and bottled water. The MPs investigated by the selected studies were classified according to their size and, as regards the bottled water, the packaging. The methodology for the identification of microparticle composition was also reported.

## 2. Materials and Methods

A focused literature search for original peer-reviewed articles on microplastic occurrence in drinking water was carried out on PubMed and Scopus (11 November 2021). The keywords searched were “microplastics”, “drinking water”, “tap water” and “bottled water” in different combinations. The research was filtered to cover a time span of 11 years, from 2010 to 2021. Furthermore, the reference lists of the identified papers were reviewed and manually searched for additional publications.

The searched literature was focused only on BW and TW (directly at the point of use). Eligible studies must have used one (or more) of the four currently validated processes for the identification of microparticle composition: Fourier-transform infrared spectroscopy (FTIR), Raman spectroscopy (RM), pyrolysis gas chromatography/mass spectrometry (Pyr-GC-MS) and scanning electron microscopy plus energy-dispersive X-ray spectroscopy (SEM-EDS). Other analytical methods used for MPs detection (visual inspection by optical microscope of stained particles, thermal decomposition methods), were excluded because they were not applied to the selected drinking water [76,77,78,79,80,81].

The use of procedural blank samples was also mandatory. Experimental studies and those not published in the English language were excluded.

The bibliographic survey was conducted separately in each electronic database by selecting the papers based on their titles and abstracts; subsequently, the full-length articles were analyzed. At the end of the search process, 21 articles (12 for tap water and 9 for bottled water) were selected for review. 

Data on the size of particles, abundance, shape, type of polymers, packaging and analytical methods used to detect MPs were extracted and reported in tables. In particular, where possible, the results were classified according to the size of the particles, whether they were small (<10 μm), medium (10–100 μm) or large (>100 μm). Furthermore, where indicated, the source of tap water, whether surface water (SW), groundwater (GW) or desalinated water (DW), and the nature of the container of bottled water, whether single-use, reusable PET (R-PET), polycarbonate track-etched (PCTE), glass or carton, was indicated. A graphical representation of the studies on MPs in tap and bottled water conducted around the world in the considered period was built by means of the software QGis (version 3.16).

## 3. Results

### 3.1. Selected Literature

MPs have been identified in freshwater bodies such as rivers, lakes and groundwater [11], raising concern for water intended for human consumption. The presence of MPs in drinking water was largely unknown and data on this issue are available only since 2016.

The interest in MPs detection in DW grew over years, as demonstrated by the increasing number of studies in this field of research (Figure 2). Over a time span of 13 years, the researchers returned 21 studies for drinking water, 12 for TW and 9 for BW. However, for both tap and bottled water, a relevant number of studies is only available from 2018, remaining the less studied field.

Figure 3 represented a geographical representation of data on MPs. MP occurrence was reported in TW in Germany [27,82,83,84], USA [82], Mexico City [85], France [82], China [86,87,88], Japan [82], Saudi Arabia [89], Norway [90], Denmark [91,92] and Finland [82]. While MP occurrence in BW was reported in waters purchased from Germany [93,94,95,96], Thailand [97], USA [85], Italy [23,98,99] and Saudi Arabia [89]. A larger study [23] was carried out on BW purchased in many countries (Indonesia, USA, India, Mexico, England, France, Germany, Italy, Brazil, Lebanon and China).

### 3.2. MPs in BW

Nine studies investigating MPs in BW were selected from the reviewed literature. They were summarized in Table 1 according to the type of packaging (PET, R-PET, Glass, Carton, PCTE), analytical method used to detect MPs, detected polymers, their shape, size and abundance as a function of particle size when they were reported.

PET was the most studied packaging in eight studies [23,89,93,94,95,97,98,99]. Two studies investigated MPs in R-PET packaging [94,95]. Glass packaging was also investigated in six studies [23,89,94,95,96,97]. Finally, carton [95] and PCTE [89] packages were each investigated in one study.

Regarding PET packaging, and according to their size, MPs smaller than 10 μm were specifically investigated by Oßmann et al. [94] and Zuccarello et al. [98], with an average concentration ranging from 2649 p/L to 5.42 × 10^7^ p/L.

MPs > 100 μm were specifically investigated only by Mason et al. [23], with an average concentration of 10.4 p/L. Five studies investigated MPs over a wider or different size range: [23] found a concentration of 315 p/L within a size range of 6.5–100 μm; [95] found a concentration of 14 ± 14 within a size range of 5–1359 μm; [99] a concentration of 148 ± 253 p/L sized ≥ 3 μm; [97], a concentration of 140 ± 19 p/L sized ≥ 6.5 μm; lastly, [89] within a size range of 25–500 μm reported a concentration of MPs from 0.9 to 4.2 p/L.

Wiesheu et al. [93] found only one MP but no information on size was available. Beyond expectations, MPs were reported for glass bottles too. Concerning sizes smaller than 10 μm, MPs were reported by Oßmann et al. [94] with an average concentration of 5864.1 p/L.

Particles larger than 100 μm were found by Mason et al. [23], with a concentration of 8.96 p/L.

Six studies investigated MPs over a wider size range, reporting the following results: the authors of [23] found an average concentration of 195 p/L sized 6.5–100 μm; in [94], a concentration of 434.1 p/L sized > 10 μm was found; in [95], 50 ± 52 p/L within a size range of 5–1359 μm; in [97], a concentration of 52 ± 4 p/L sized ≥ 6.5 μm and in [96], a concentration of 317 ± 257 p/L withing a size range of (11–500 μm); Almaiman et al. [89] detected no particles in analyzed samples. Only two studies [94,95], investigated MP presence in R-PET packaging. According to their size, MPs smaller than 10 μm were detected only by Oßmann et al. [94] with an average concentration of 4805.9 p/L, while for particles > 10 μm, a concentration of 83.1 p/L was reported. By contrast, Schymanski et al. [95] reported an average concentration of 118 ± 88 p/L size from 5 to 1359 μm. One study [95] investigated MP presence in carton packaging reporting an average concentration of 11 ± 8 p/L.

PCTE packaging was investigated by Almaiman et al. [89] but no particles were detected. 

In addition, four studies [94,95,98,99] reported the concentration in mass of MPs for PET bottles and it was 0.1 μg/L, 1.8 μg/L, 1.71 μg/L and 656.8 μg/L ± 632.9, respectively. Regarding the chemical composition of particles, PET and PE were the most abundant polymers reported in all papers. PP was reported in four studies [23,89,94,95,97]. PS was another common polymer found through three surveys [23,89,96]. PA or NY was reported by [23,89,95,97]. Whereas, less common was PU, which was found by [89] in 2021. Additive presence was also investigated by Oßmann et al. (pigments and antioxidant additives) in 2018 [94] and by Winkler et al. (lubricants) in 2019 [99].

Some studies reported information about MP shape. Fragments seemed to represent the predominant particle shape [23,94,96] followed by fibers [93,97]. Other shapes (films, pellets, foam) were reported by [23,94].

### 3.3. MPs in TW

Further, 12 studies investigating MPs in TW were obtained from the reviewed literature. They are summarized in Table 2, according to the water source, analytical method used to detect MPs, detected polymers, their shape, size, and abundance as a function of particle size, when it was reported.

Groundwater was the main source of tap water supply in the selected papers [27,82,83,84,85,88,92]. Three studies [82,87,90] considered tap water fed from surface sources and one [89] from desalinated water. In all cases, samples were taken at the point of use of the water (houses, public fountains, etc.).

According to their size, MPs smaller than 10 µm were specifically detected only by Shen et al. [87], with an average of 266 ± 56 p/L. Four studies [82,87,91,92] investigated MPs sized 10–100 µm and found an average concentration ranged from 0.2 ± 0.1 to 63 ± 11 p/L. Five studies [82,85,87,91,92] specifically investigated the occurrence of large MPs (>100 μm) reporting an average concentration ranging from no detected particles to 18 ± 7 p/L. Five studies investigated MPs over a very wide range of sizes and reported the overall result: [84] found a concentration of 40 ± 48 p/m3 of MPs with a size 5–1000 μm, [86] a concentration of 440 ± 275 p/L of MPs with a size 1–5000 μm, [27] a concentration of 0.7 p/m3 of MPs with a size 50–150 μm, [88] a concentration of 13.23 p/L of MPs with sizes from <50 μm to >200 μm [89] performed their analysis in two tap water samples detecting in one of them a concentration of 1.8 p/L. Finally, the authors of [90] and [83] detected no particles in their samples. In general, the concentration of particles in tap water increased as their size decreased. Regarding polymeric composition, PP and PE, were each documented in six studies [27,82,84,86,87,89,91,92]. PET and PS were both documented in five studies [84,86,87,91,92]. PVC was reported in three studies [27,82,87]. In some cases [27,85], epoxy resin was detected. It is a coating material utilized to avoid corrosion and is probably derived from the plastic container for water storage.

Mukotaka et al. [82] detected several types of MPs in tap water samples from Japan, the European Union and the United States, finding overall mean concentrations of 29 ± 45 p/L, 66 ± 37 p/L and 46 ± 32 p/L, respectively.

## 4. Discussion

Modern society is inevitably dependent on plastic polymers so much that, to date, plastics represent the greatest pollutants worldwide. MP presence was documented in all environmental matrices, including water supplies intended for human consumption (i.e., rivers, lakes or groundwater). Humans are exposed to them by dermal contact, air inhalation and food ingestion [37]. Recently, increasing literature has confirmed MP occurrence in drinking water also, raising significant concerns for human health due to the intensity and duration of exposure even by vulnerable subjects. Although only a few studies investigated particles smaller than 10 μm [94,98], this review highlighted that MP abundance increased as the particle size decreased both in TW and BW. Furthermore, MP concentration was higher in BW than in TW, raising much more concern for public health because of the prevalent consumption of BW as a source of DW [100,101]. Some studies displayed a relatively low MP concentration in TW when compared to untreated water, probably due to a good removal efficiency (up to 90%) in water treatment plants [87,102,103]. Others reported MP contamination in TW originating from the air and water pipes [11,36,82].

When considering BW, MP concentration was higher in PET packaging than in others, highlighting that this packaging represents a source of plastic pollution. In this regard, the authors of [23] reported a different abundance of MPs (204 p/L vs 1410 p/L, for glass and PET bottles respectively) after having analyzed the same water bottled in different packaging. Plastic items, present even in glass bottles (i.e., plastic layers under the cap), could also release small plastic particles into BW. Winkler et al. [99] demonstrated that a large contribution of MP contamination in BW is derived from cup-neck pressures (after opening them more than 100 times) with respect to the mechanical stress to which bottles are exposed during normal daily use (drinking, carrying and by handling bottles). During their whole life cycle, bottles are subjected to different pressures such as squeezing, cleaning, labeling, transportation, storage, UV exposure and temperatures which may influence MP contamination.

In this regard, in order to limit MPs released into bottled water, abrasion and resistance test should be performed on cap sealing materials. Since bottles caps are considered the main contributors of MPs released, it would be better to invest in alternative solutions to replace plastic caps as proposed by Jadhav et al. [104] (i.e., edible bioplastics such as chitosan-based nanocomposite, milk plastic, etc.). Moreover, innovative packaging technologies to unscrew the bottle caps in other ways, such as the “easy to open caps”, could be a possible solution to minimize the release of particles. However, further studies are needed to confirm this hypothesis.

Since the presence of MPs was often described in glass, carton or PCTE packaged water, the contamination could occur during bottling processes (filling, capping, etc.) [96]. As suggested by the authors of [94] and [97], processes occurring in bottling plants play a fundamental role in terms of MP contamination. 

Considering PET packaging, single-use PET showed a lower MP abundance compared to R-PET. Oßmann et al. [94] emphasized this aspect by comparing newish versus older R-PET packaging. The latter showed the highest MP concentration probably caused by aging of the bottle material. 

Regarding polymeric composition, PET and PE were the most common polymers found in BW. Presumably, PET is derived from the packaging itself and the bottleneck, while PE is derived from the cap of which it is the main constituent [99]. For TW, PP and PE were the common polymers found. Further polymers, such as epoxy resins and PVC, were also reported by other studies suggesting leaching of the water pipes of the supply systems or storage tanks. MP concentrations in the reviewed studies differ consistently. This variability could be associated with different factors. First of all, there are intrinsic factors such as contamination present at the source, distribution systems for water supplied by public aqueducts, etc. Besides these, extrinsic factors play a crucial role in the abundance of MPs. 

In this regard, the analytical procedures used for chemical characterization (Figure 4) are of paramount importance and represent a discriminating factor both in terms of the size of particles detected and MP abundance. MPs in TW were lower in abundance and higher in size compared to BW, indeed, TW was usually analyzed by FTIR spectroscopy, so the number and size of MPs can be underestimated due to the instrumental incapability of detecting MPs smaller than 10 μm [105]. Whereas BW was also analyzed by RM spectroscopy or other validated methods such as SEM/EDS, which are capable of detecting particles smaller than 10 μm. Therefore, the use of different analytical procedures among studies emphasized the diversity of MP abundance between and within the two drinking waters, making the comparison difficult.

Studies of drinking water are mainly related to particles size > 1 µm, while the smallest particles (<1µm) are posing a challenging task due to the analytical detection limits [105,106].

Regarding the implication for human health, the occurrence of MPs in DW is an alarming issue, considering the direct and long-term exposure of the entire population, including the most vulnerable groups. The World Health Organization in 2019 [107], based on previous studies, indicated a low health concern for humans via DW, stating that there is no evidence of effects on humans related to MPs exposure. However, there is an open debate on this issue among the scientific communities, in light of the latest evidence [108,109]. In fact, toxicity seemed to be low or not relevant at lower MP concentrations and for larger particles. Instead, experimental studies on animal models and human cell lines showed significant toxic effects for smaller particles [65,110].

The possible effects of MPs could depend on the time and intensity of exposure as well as the susceptibility of the host. In this regard, the concentration of the mass of MPs per volume of beverage is an important parameter for risk assessment, which, however, was reported only in four studies [94,95,98,99].

Considering data reported by previous studies on both BW and TW consumption, the authors of [111] estimated an average annual intake of approximately 78,000 MPs per year for male children, 133,000 MPs per year for male adults, 67,000 MPs per year for female children and 97,000 MPs per year for female adults.

Zuccarello et al. [98], analyzing particles down to 0.5 μm in size in BW, reported an Estimated Daily Intake (EDI) of MPs for adults and children of 1,531,524 p/kg/ body-weight/day corresponding to 40.1 μg/kg body-weight/day and 3,350,208 p/kg/ body-weight/day corresponding to 87.8 μg/kg/body-weight/day, respectively.

The exposure to MPs hides an indirect effect related to the presence of other substances, such as additives, considered biologically and toxicologically relevant for humans, that are present in plastic products and which may be leached out from the particles themselves. Many studies reported the presence of additives in DW such as epoxy resins, antioxidants, lubricants, phthalates and pigments [94,97,112,113]. They might have originated from the leaching of the plastic packing for BW or plastic pipes in drinking water networks for TW, due to aging, UV exposure, rising temperatures and other external forces [113]. MPs can be also considered a vehicle for other toxic substances or microorganisms, which may adhere to the surface and be conveyed into the human body.

## 5. Conclusions

The reduction of human exposure to MPs represents a challenge for the years to come. MPs were well-documented both in tap and bottled water, determining a global long-term exposure. The main considerations of the finding of this research are the following:The reviewed studies showed that BW was more contaminated than TW. In particular, MPs were found to be more abundant in R-PET and glass bottles indicating a relay from the packaging or contamination from bottling processes, respectively. This underlines the need for a new direction in the production of innovative materials to be used for packaging in order to reduce the release of MPs in BW.The presence of MPs in TW, although lower than BW, suggests environmental contamination both before and after treatments. However, the lower concentration of MPs in TW than in natural sources indicates a high removal rate of MPs in drinking water treatment plants. This evidence should encourage consumers to drink TW instead of BW, in order to limit their exposure to MPs and produce less plastic waste.The high variability in the results makes it difficult to compare the findings of different studies and build up a general hypothesis on human health risk. A globally shared protocol is needed in order to harmonize results also in view of the monitoring plans for the emerging contaminants, including MPs, introduced by the new European regulation (EU Directive, 2020/2184) [114].

## Figures and Tables

**Figure 1 ijerph-19-05283-f001:**
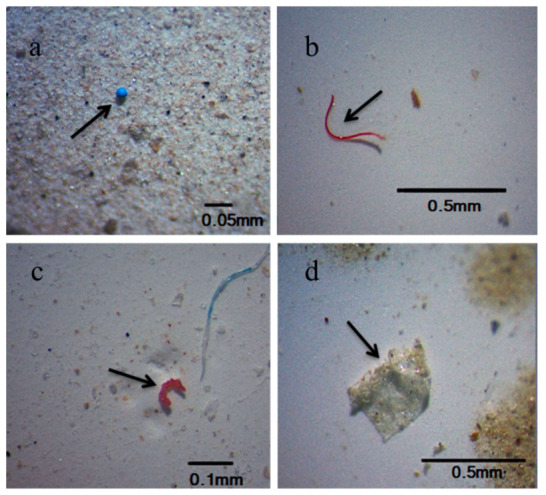
Types of MPs in the surface water and sediments from coastal Guangdong (**a**–**d**): (**a**) pellets, (**b**) fibers, (**c**) fragments, (**d**) films [12].

**Figure 2 ijerph-19-05283-f002:**
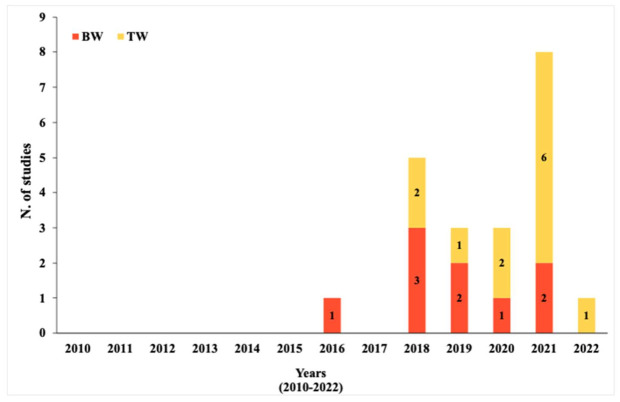
Temporal representation of current studies on DW from 2010 to 2022.

**Figure 3 ijerph-19-05283-f003:**
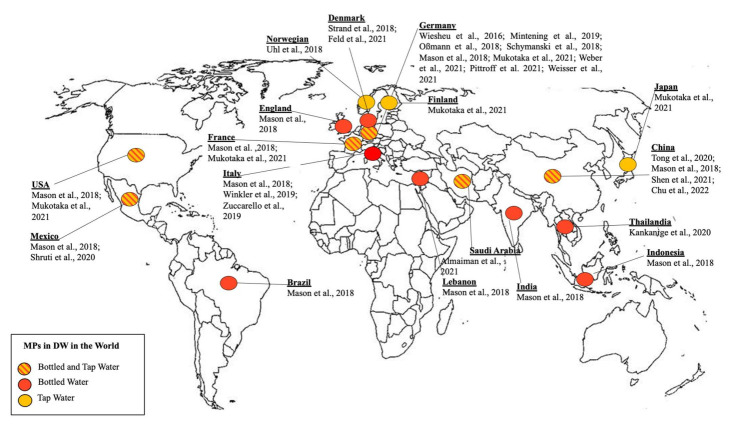
Geographical representation of MPs data in drinking water. TW: Tap Water; BW: bottled water.

**Figure 4 ijerph-19-05283-f004:**
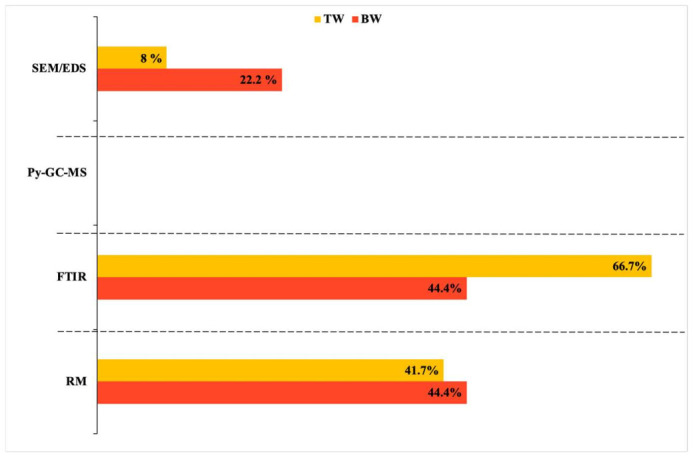
Validated analytical procedures and percentage of application in DW studies. RM: Raman Spectroscopy; SEM/EDS: Scanning Electron Microscopy/Energy Dispersive X-ray spectrometry; FTIR: Fourier—Transform Infra-red spectroscopy; Py-GC-MS: Pyrolysis Gas Chromatography Mass Spectrometry.

**Table 1 ijerph-19-05283-t001:** Analytical methods, characteristics, and abundance of MPs detected in bottled water, PET (single-use and reusable), glass and carton. PE: polyethylene; PP: polypropylene; PS: polystyrene; PET: polyethylene terephthalate; PEST (PES—polyether sulfone + PET); PA: polyamide; PBN: 1-pyrenebutyric acid N-hydroxysuccinimidyl ester; PU: polyurethane; PCTE: polycarbonate track-etched.

Reference	Packaging	Analytical Method	Shape	Detected Polymers	Particle Size	Abundance
Wiesheu et al. [93]	PET	RM	Fibers	PET	-	1 particle
Mason et al. [23]	PET, Glass	FTIR	Fragments, Film, Fiber, Foam, Pellet	PP, NY, PS, PE, PEST	6.5–100 μm; >100 μm	315 p/L (PET); 195 p/L (Glass) 10.4 p/L (PET); 8.96 p/L (Glass)
Obmann et al. [94]	PET, R-PET, Glass	RM	Fragments Fibers Films	PET, PE, PP	1–10 μm; >10 μm	2649 p/L (PET) 4805.9 p/L (R-PET) 5864.1 p/L (Glass) 83.1 p/L (R-PET) 434.1 p/L (Glass)
Schymanski et al. [95]	PET, R-PET, Glass, Carton	RM	-	PEST, PE, PP, PA, others	5 μm–1359 μm	14 ± 14 p/L (PET) 118 ± 88 p/L (R-PET) 50 ± 52 p/L (Glass) 11 ± 8 p/L (Carton)
Winkler et al. [99]	PET	SEM-EDS	-	PET, PE	≥3 μm	148 ± 253 p/L
Zuccarello et al. [98]	PET	SEM-EDS	-	-	0.5–10 μm	5.42 × 10^7^ p/L
Kankanige et al. [97]	PET, Glass	FTIR RM	Fibers, Fragments	PET, PE, PP, PA	≥6.5 μm	140 ± 19 p/L (PET) 52 ± 4 p/L (Glass)
Almaiman et al. [89]	PET, Glass, PCTE,	FTIR	-	PE, PS, PET, PP, PA, PU	25–500 μm	From 0.99 to 4.2 p/L (PET) <LOQ (Glass) <LOQ (PC)
Weisser et al. [96] 2021	Glass	FTIR	Fragments	PE, PS	11–500 μm	317 ± 257 p/L

**Table 2 ijerph-19-05283-t002:** Analytical methods, characteristics, and abundance of MPs detected in TW and DWTP. PET: polyethylene terephthalate; PP: polypropylene; PE: polyethylene; PS: polystyrene; PVC: polyvinyl chloride; PA: polyamide; PAM: polyacrylamide; PPS: polyphenylene sulfide; PEST: (PES − polyether sulfone + PET); SW: surface water; GW: groundwater; DW: drinking water.

Reference	Water Source	Analytical Method	Shape	Detected Polymers	Particle Size	Abundance
Strand et al. [91]	-	FTIR	Fibers, Fragments, Films	PET, PP, PS, others,	10–100 μm; >100 μm	0.3 p/L (10–100 μm); <LOD (>100 μm);
Uhl et al. [90]	SW	FTIR	-	-	-	<LOQ
Mintening et al. [27]	GW	FTIR	Fibers	PEST, PVC, PE, PA, Epoxy resin	50–150 μm	0.7 p/m^3^
Shruti et al. [85]	GW	SEM-EDS, RM	Fibers	PTT and Epoxy resin	>100 μm	18 ± 7 p/L
Tong et al. [86]	-	RM	Fragments, Fibers, Spheres	PE, PP, PE + PP, PPS, PS, PET, others	1–5000 μm	440 ± 275 p/L
Weber et al. [83]	GW	RM	-	-	-	<LOQ
Shen et al. [87]	SW	SEM, FTIR, RM	Fragments, Fibers, Spheres	PA, PVC, PP, PET, PE, Others	1–10 μm; 10–100 μm; >100μm	266 ± 56 p/L 63 ± 11 p/L 14 ± 5 p/L
Pittroff et al. [84]	GW	RM	-	PE, PET, PP, PA	5–1000 μm;	40 ± 48 p/m^3^
Feld et al. [92]	GW	FTIR	Fragments, Fibers	PP, PS, PET, others	10–100 μm; >100 μm	0.2 ± 0.1 p/L 0.31 ± 0.14 p/L
Almaiman et al. [89]	DS	FTIR	-	PE	25–500 μm	1.8 p/L (1 of 2 samples)
Mukotaka et al. [82]	GW-SW	FTIR	Fragments, Spheres Fibers	PS, SEBS, PP, PES, PE, PVC, others	10–100 μm; >100 μm	32 ± 29 p/L 7.3 ± 9.1 p/L
Chu et al. [88] 2022	GW	FTIR	Fragments, Fibers	PEST, NY, PS	>10 μm	13.23 p/L

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
