# Peer review of "Occurrence of Microplastics in Tap and Bottled Water: Current Knowledge"

_ijerph, 2022, doi:10.3390/ijerph19095283_

Round 1

Reviewer 1 Report

The manuscript concerns occurrence of microplastics in tap and bottled water: current knowledges and human health implications. The reviewed studies (n=21) showed the presence of microplastics (MPs) in tap (TW) and bottled (BW) water, increasing concerns for public health due to the possible toxicity associated with their polymeric composition, additives and other compounds or microorganism adsorbed on their surface. MPs concentration increase by decreasing particles size and was higher in BW than in TW. Among BW, reusable PET and glass bottles showed a higher MPs contamination than other packages. The lower MPs abundance in TW than in natural sources indicates a high removal rate of MPs in drinking water treatment plants. Remarks: The review's significant achievements and recommendations should be highlighted in the conclusion. What are the lessons from this research? It should be added in a more detailed way.

Author Response

Reviewer 1

The manuscript concerns occurrence of microplastics in tap and bottled water: current knowledges and human health implications. The reviewed studies (n=21) showed the presence of microplastics (MPs) in tap (TW) and bottled (BW) water, increasing concerns for public health due to the possible toxicity associated with their polymeric composition, additives and other compounds or microorganism adsorbed on their surface. MPs concentration increase by decreasing particles size and was higher in BW than in TW. Among BW, reusable PET and glass bottles showed a higher MPs contamination than other packages. The lower MPs abundance in TW than in natural sources indicates a high removal rate of MPs in drinking water treatment plants. Remarks: The review's significant achievements and recommendations should be highlighted in the conclusion. What are the lessons from this research? It should be added in a more detailed way.

Answer to Reviewer 1

We thank the Reviewer for his valuable comments which are helpful for revising the manuscript. In the conclusion section we reported the main findings of our review highlighting that bottled water was more contaminated by MPs than tap water. In particular, R-PET and glass bottles resulted the most contaminated and, for this reason, we encouraged the consumption of TW respect of BW, both to limit the exposure to MPs and to reduce plastic waste. We underlined the need of a new direction in the production of innovative materials to be used for packaging to minimize the release of MPs in water [lines 454-456], with some examples in the discussion section [lines 362-368]. Finally, we pointed the requirement of a shared protocol in order to harmonize results of different studies.

Reviewer 2 Report

The manuscript entitled “Occurrence of Microplastics in Tap and Bottled Water: Current Knowledges and Human Health Implications” reported the presence of microplastics (MPs) in tap (TW) and bottled (BW) water has raised concerns about their potential toxicity owing to their polymeric nature, additives, and other substances or microorganisms adsorbed on their surface. Based on the quality of this work, the manuscript can be considered for publication after clarifying some points. The main comments are as follows:

1. The authors should show some images of microplastic to an easy understanding of the size and morphology of microplastic. 
2. The authors should discuss currents solutions or technologies to solve the microplastic problem. 
3. The authors should add the information on chemical properties of MPs.
4. The authors should discuss and give some examples of some health problems caused by MPs. 
5. Please carefully check the manuscript such as the abbreviation of MPs. 

Author Response

Answer to Reviewer 2

The manuscript entitled “Occurrence of Microplastics in Tap and Bottled Water: Current Knowledges and Human Health Implications” reported the presence of microplastics (MPs) in tap (TW) and bottled (BW) water has raised concerns about their potential toxicity owing to their polymeric nature, additives, and other substances or microorganisms adsorbed on their surface. Based on the quality of this work, the manuscript can be considered for publication after clarifying some points.

We thank the Reviewer for his revision of the manuscript and his considerations which allowed us to clarify important issues. All the comments have been accepted as indicated below.

The main comments are as follows:

1.The authors should show some images of microplastic to an easy understanding of the size and morphology of microplastic.

[Line 62-63] We have added a picture which highlight the main microplastics morphology and size.

  1. The authors should discuss currents solutions or technologies to solve the microplastic problem.

[lines 362-368] we reported some solutions and directions that research in the field of innovation should pursue in order to minimize particles released into water, such as abrasion and resistance test for cap sealing and the possibility of a new form of caps such as the “easy to open caps” as a possible solution to reduce the released of particles.

  1. The authors should add the information on chemical properties of MPs.

[Line 70-74] We provided more information on the polymeric nature of the most commonly detected MPs.

  1. The authors should discuss and give some examples of some health problems caused by MPs.

[Lines 166-178] We discussed the possible human effects. As there are no in vivo studies, therefore, the direct effects of MPs on humans are still unclear and largely unknown. Some studies detected the presence of MPs in placenta tissue, blood, etc. However, no data on health problems caused by MPs were documented and only possible effects on the basis of in vitro studies on human cell lines and on animal models can be hypothesized.

  1. Please carefully check the manuscript such as the abbreviation of MPs.

MPs abbreviations were checked throughout the text and corrected.

Reviewer 3 Report

The present review article addresses a topic of great importance nowadays: the implications of microplastics in Human Health.  The idea of the authors is interesting, review and compile information of the different papers about this theme. However, in my opinion is necessary more state-of-the-art research, with more recent publications.

The estimative made by the authors, the cumulative plastics presented in the line 37, should be made for present scenario (2022), they should present more recent results. Also, the explanation of the size of the microplastics, as well as their degradation, seem to be very poor and confusing; here I advise some studies such as:

  • Microplastics in Ecosystems: From Current Trends to Bio-Based Removal Strategies (doi:10.3390/molecules25173954)
  • On some physical and dynamical properties of microplastic particles in marine environment (doi: 10.1016/j.marpolbul.2016.04.048)
  • The chemical behaviors of microplastics in marine environment: A review (doi: 10.1016/j.marpolbul.2019.03.019)
  • The physical impacts of microplastics on marine organisms: A review (doi: 10.1016/j.envpol.2013.02.031)
  • Microplastics: Finding a consensus on the definition (doi: 10.1016/j.marpolbul.2018.11.022)
  • Occurrence of microplastics in commercial fish from a natural estuarine environment (doi: 10.1016/j.marpolbul.2018.01.044)

Similarly, the techniques used in the characterization of microplastics, could also be more explored. Some suggestions:

  • A new approach for routine quantification of microplastics using Nile Red and automated software (MP-VAT) (doi: 10.1016/j.scitotenv.2019.07.060)
  • A Critical Review of Extraction and Identification Methods of Microplastics in Wastewater and Drinking Water (doi: 10.1021/acs.est.9b06672)
  • Identification of microplastics using 4-dimethylamino40-nitrostilbene solvatochromic fluorescence (doi: 10.1002/jemt.23841)
  • Microplastics in freshwater systems: A review on occurrence, environmental effects, and methods for microplastics detection (doi: 10.1016/j.watres.2017.12.056)
  • Optimization, performance, and application of a pyrolysis-GC/MS method for the identification of microplastics (doi:10.1007/s00216-018-1279-0)
  • Análise de microplásticos de polietileno em amostras ambientais, utilizando-se um método de decomposição térmica (doi: 10.1016/j.watres.2015.09.002)

Finally, I think the focus in the title is not very evident throughout the text of the article. At this point, I think it would be interesting to highlight the harmful effects of MPS on human health.

Author Response

Answer to Reviewer 3

The present review article addresses a topic of great importance nowadays: the implications of microplastics in Human Health. The idea of the authors is interesting, review and compile information of the different papers about this theme. However, in my opinion is necessary more state-of-the-art research, with more recent publications.

We thank the Reviewer for the useful suggestions which improved our paper. We have provided the amendments as suggested.

The estimative made by the authors, the cumulative plastics presented in the line 37, should be made for present scenario (2022). They should present more recent results.

[Line 37-39] We uploaded with recent existent data on plastic and waste production as reported by the latest version of Plastic Europe [2021].

Also, the explanation of the size of the microplastics, as well as their degradation, seem to be very poor and confusing; here I advise some studies such as:

  • Microplastics in Ecosystems: From Current Trends to Bio-Based Removal Strategies (doi:10.3390/molecules25173954)
  • On some physical and dynamical properties of microplastic particles in marine environment (doi: 10.1016/j.marpolbul.2016.04.048)
  • The chemical behaviors of microplastics in marine environment: A review (doi: 10.1016/j.marpolbul.2019.03.019)
  • The physical impacts of microplastics on marine organisms: A review (doi: 10.1016/j.envpol.2013.02.031)
  • Microplastics: Finding a consensus on the definition (doi: 10.1016/j.marpolbul.2018.11.022)
  • Occurrence of microplastics in commercial fish from a natural estuarine environment (doi: 10.1016/j.marpolbul.2018.01.044)

[Lines 42-62] The explanation of microplastics size and their degradation was improved following the suggested literature.

Similarly, the techniques used in the characterization of microplastics, could also be more explored. Some suggestions:

  • A new approach for routine quantification of microplastics using Nile Red and automated software (MP-VAT) (doi: 10.1016/j.scitotenv.2019.07.060)
  • A Critical Review of Extraction and Identification Methods of Microplastics in Wastewater and Drinking Water (doi: 10.1021/acs.est.9b06672)
  • Identification of microplastics using 4-dimethylamino40-nitrostilbene solvatochromic fluorescence (doi: 10.1002/jemt.23841)
  • Microplastics in freshwater systems: A review on occurrence, environmental effects, and methods for microplastics detection (doi: 10.1016/j.watres.2017.12.056)
  • Optimization, performance, and application of a pyrolysis-GC/MS method for the identification of microplastics (doi:10.1007/s00216-018-1279-0)
  • Análise de microplásticos de polietileno em amostras ambientais, utilizando-se um método de decomposição térmica (doi: 10.1016/j.watres.2015.09.002)

In lines [201-204] we have also reported the other analytical methods used for MPs analysis (within those suggested). These methodologies are not applied to the MPs detection in drinking water and also present some limitations about the chemical characterization of particles. So that, we selected the four validated technique reported in the text.

Finally, I think the focus in the title is not very evident throughout the text of the article. At this point, I think it would be interesting to highlight the harmful effects of MPS on human health.

Thank you for your suggestion. We have added more consideration about the possible effects on human health [lines 166-178]. It should be highlighted that, although the presence of MPs was detected in human tissues such as placenta, blood, etc., the direct effects of MPs on humans are still unclear and largely unknown.  Possible harmful effects on humans can be hypothesized on the basis of in vitro studies on human cell lines and animal models, which have been more deeply discussed as suggested. In any case, since there is not enough evidence about the effects on human health and the aim of the review was focused on the presence of MPs in drinking water, we preferred to delete “Human health implications” from the title.

Round 2

Reviewer 3 Report

The manuscript has been revised according to the comments.